# Physics-Based Modeling and Fluttering Dynamic Process Simulation for Catkins

Jiaxiu Zhang [1], Meng Yang [1,2,*], Benye Xi [3], Jie Duan [3], Qingqing Huang [4] and Weiliang Meng [5,6]

1   School of Information Science and Technology, Beijing Forestry University, Beijing 100083, China; jiaxiuzhang@bjfu.edu.cn
2   Engineering Research Center for Forestry-Oriented Intelligent Information Processing of National Forestry and Grassland Administration, Beijing 100083, China
3   College of Forestry of Beijing Forestry University, Beijing Forestry University, Beijing 100083, China; benyexi@bjfu.edu.cn (B.X.); duanjie@bjfu.edu.cn (J.D.)
4   School of Technology, Beijing Forestry University, Beijing 100083, China; huangqingqing@bjfu.edu.cn
5   State Key Laboratory of Multimodal Artificial Intelligence Systems, Institute of Automation, Chinese Academy of Sciences, Beijing 100190, China; weiliang.meng@ia.ac.cn
6   School of Artificial Intelligence, University of Chinese Academy of Sciences, Beijing 100049, China
*   Correspondence: yangmeng@bjfu.edu.cn

**Abstract:** Flight simulation of catkins using computer technology helps their prevention and control. However, this is a challenging task due to the complex characteristics, and irregular shapes of catkins, while existing methods mainly focus on rain and snow, which are not suitable for catkins. In this paper, we propose a physics-based algorithm for the dynamic simulation of fluttering catkins. Our approach includes an L-system based 3D modeling method for simulating the natural phenomena of the catkin. We consider the motion of wind, free fall of catkins, and the dynamics of catkins under the joint action of attraction between them, while adhering to the physical motion law of catkins. To provide wind force, we first establish a three-dimensional wind field based on Boltzmann's equation. We then use the kernel function idea to calculate the attraction force between catkins and finally update the position of the catkin. We incorporate the phenomena of collision and adhesion, attraction, and accumulation of catkins while simulating motion states depending on the adjusted wall height and ground humidity parameters. Our approach overcomes limitations of previous models by achieving good simulation while using relatively less code to simulate various realistic motion states. According to our users' study, more than 71% of users found the simulation results to be acceptable, authentic, and realistic, confirming the authenticity of our simulation. Our method can generate highly realistic effects, significantly improving efficiency by several orders of magnitude compared to manual modeling. In addition, it can effectively simulate the dynamics of catkins in different scales, providing a decision-making reference for catkin control.

**Keywords:** clustering; catkin modeling; wind field simulation; catkin control

## 1. Introduction

Since the 1960s and 1970s, various departments and regions in China have been actively promoting urban reforestation by planting a large number of poplar and willow trees, which have significantly improved the urban ecological environment and helped form distinct urban landscapes with notable ecological benefits. However, the catkins produced by poplar or willow trees have created some unavoidable problems. In recent years, during the reproductive maturity stage of female poplar or willow trees, they have produced a significant number of catkins in spring. This phenomenon poses a serious threat in densely populated urban areas and other regions with special needs. The concentrated presence of female poplar or willow trees results in excessive catkin flocculation, which negatively impacts air quality, human health, traffic safety, and public security, interferes

with normal industrial production and scientific research activities, and seriously affects ecological environment construction and the creation of livable cities. The detrimental influence of catkin fluttering on the environment and on the progress of creating sustainable urban spaces is apparent and substantial.

The catkin fluttering problem has direct implications for people's well-being, and therefore effective management is highly demanded by society. Despite the Chinese government's implementation of various measures to mitigate the impact of catkins on daily life, the sheer abundance of trees has posed challenges in achieving substantial results. In the past, predicting the processes and outcomes related to catkins relied on observation data, which took several years, and differed depending on the area and environment, sometimes requiring multiple observations to obtain data, resulting in inefficiencies. Consequently, computer simulation technology has become indispensable for achieving effective catkin control. The increased development of computer graphics and image technology has led to the widespread application of visualization systems, making simulation technology a powerful tool for analyzing situations and developing effective plans. Significant advancements have been made in realistically simulating natural phenomena, like rain [1], snow [2], smoke [3], and fire [4], while these developments have served as inspiration for our efforts in simulating catkins. Despite the abundance of poplar and willow trees in nature, few people understand their morphology and structure. Therefore, computer-aided 3D modeling of catkins can assist researchers in comprehending the catkin's properties and structure. Furthermore, simulating catkin floating, collision and adhesion, attraction, and accumulation processes through computer technology can help anticipate the extent and degree of problems relating to catkins, providing an effective reference and decision-making basis for catkin control.

In the past, many researchers have attempted to analyze the complex flow of organisms such as pollen, which has also inspired the research in this paper. Mazinani et al. [5] conducted fluid dynamics (CFD) simulations in Ansys Fluent [6] software to predict pollen motion flow, while Alaa et al. [7] established a CFD model to calculate the resistance of different fungal spores and pollen grains, in order to understand the relationship between their fluidity and pathogenicity. Further, Cresswell et al. [8] used CFD and wind tunnel experiments to study the aerodynamic interaction between flowers and pollen suspensions. However, the above research only focuses on organisms such as pollen and cannot fully migrate into the simulation of the flight process of catkins. Catkins have special shapes and lightweight characteristics, and their flight process may be influenced by factors such as air flow, gravity, and the shape and mass of the catkins themselves. Therefore, simulation research on catkins' flight still faces challenges, such as low visual fidelity and insufficient accuracy of results.

Simulating catkin fluttering is challenging due to the influence of various factors, such as wind, temperature, gravity, and the unique morphological structure and physical properties of catkins. Moreover, the appearance and motion trajectory of fluttering catkins in the real world lacks explicit rules. In this paper, we propose a physics-based modeling and fluttering dynamic process simulation algorithm for catkins in Figure 1. In our method, we establish a three-dimensional model and simulate the natural phenomenon of catkin fluttering by employing aerodynamics according to the physical principles of catkins. In the realm of simulating wind fields, two general methods exist. One approach involves assuming the fluid's continuity in both time and space, as exemplified by the N-S equation [2]. However, due to its nature as a nonlinear system, solving it proves challenging, hindering the dynamic simulation of wind interactions with willow catkins in the wind field. An alternative method is rooted in the assumption of continuous media, and employs the Boltzmann method grounded in physical statistics [9], which imparts deeper physical information compared to the N-S equation. Hence, we select the Boltzmann equation to construct a three-dimensional wind field model. This article mainly focuses on the macroscopic characteristics of the movement trajectory and velocity distribution of catkins in the wind field, without paying much attention to the details of turbulent

phenomena. Therefore, the D3Q15 model, which can provide the overall motion of the fluid and has high computational efficiency, is selected for simulation. Our simulation algorithm allows for a visual and intuitive understanding of the structural form of catkins and their fluttering behavior in the wind. It also facilitates further investigation into the fundamental nature of catkins and enables the simulation of their adhesion and accumulation under different conditions.

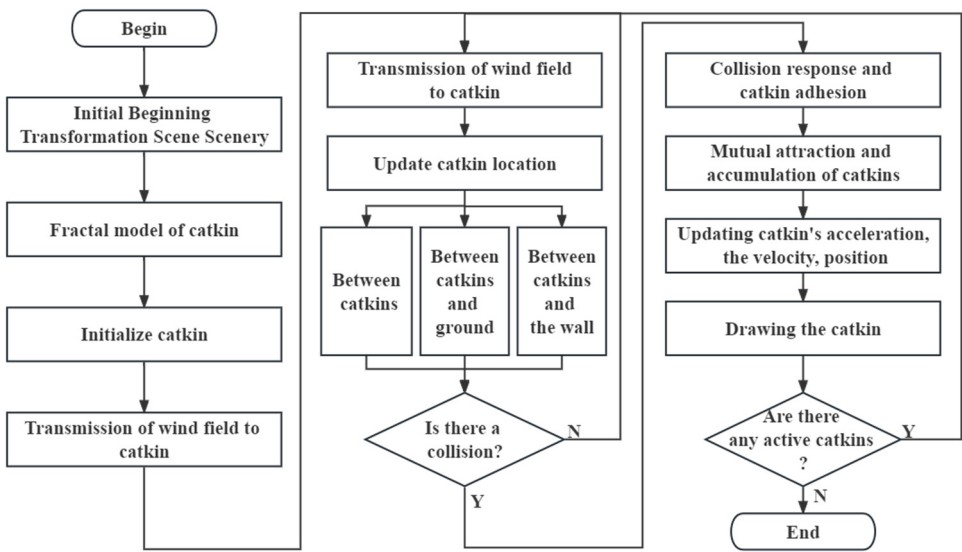

**Figure 1.** Flowchart of physics-based modeling and fluttering dynamic process simulation algorithm for fluttering catkins.

Our simulation of fluttering catkins includes static and motion patterns, and the main contributions are as follows:

- The drawing of the catkin is achieved by combining the natural form and physical characteristics to create the catkin system, including fractal modeling, initialization, and updating.
- The influence of the external environment on the motion of the catkin is taken into account. We meticulously analyze the force situation during the fluttering process, encompassing the motion of the catkin as the motion of wind and the free fall of the catkin, as well as the integrated motion under the effect of attraction between catkins, in order to build the motion model. We construct a motion model based on the principles derived from the Boltzmann equation. The wind field simulation updates the position, velocity, and acceleration of catkins in real-time as the wind field changes. This allows us to depict the fluctuating motion characteristics of catkins during the fluttering process.
- The collision and adhesion of catkins are simulated, including collision between catkins, collision between catkins and the ground or wall, and the adhesion of catkins after collision.
- The attraction and accumulation of catkins are simulated, in which the attraction between catkins is calculated using the Radial Basis Function (RBF) to realize the attraction of catkin clusters to individual clusters, and the accumulation phenomenon is simulated by the multiple attraction between catkins.
- The purpose of this paper is to simulate the flying process of catkins in nature, including the modeling of catkins, and of phenomena such as swaying, collision, adhesion, attraction, and accumulation of catkins under wind force, in order to provide guidance for the prevention and control of catkins. We conduct our research based on the following assumptions. (i) The influence of resistance on their movement is ignored, due to the small volume, high density, and light weight of catkins; (ii) The flight process of catkins is considered stable and primarily affected by the wind stroke in the wind field.

(iii) The collision between catkins and ground or wall obeys the law of conservation of momentum, without considering the deformation of catkins.

- Our method excels in conducting detailed simulations of both the static and dynamic aspects of the flying process of catkins, resulting in a realistic user experience. By simulating the flying process of catkins using computer technology, we can accurately predict their flight paths, assisting urban managers in implementing necessary cleaning and greening measures. Additionally, the simulation results enable real-time alerts for catkin outbreaks, facilitating timely preventive actions. These results provide a scientific basis for government planning departments and urban planners, effectively reducing the negative impact of catkins on the urban environment.

## 2. Related Work

### 2.1. Modeling of Natural Scenery

With the rapid development of computer graphics and image technology, the application of visualization systems has become ubiquitous, with people increasingly demanding high levels of simulation fidelity. The key to achieving this is the modeling of natural objects, which plays a crucial role in simulating realism. In fact, the distinctive static fractal characteristics of plants primarily manifest in their topological structures and geometric forms [10]: plant components are structured in a self-similar manner to the whole, such as in the bifurcation structure of branches and roots, while the external contours of each component structure are statistically self-similar to the plant as a whole. For instance, the shapes of organs like leaves and fruits correspond to their respective branches or parts of the overall plant.

Numerous virtual plant modeling methods have been proposed. One method is the L-system [11], which provides a framework for constructing morphological models of fractal virtual plants. However, it can be difficult to draw coloring and textures with L-systems. Another method is the iterative function system fractal (IFS) [12], which uses different probabilities for fractal growth. The diffusion-limited coalescence (DLA) model [13] focuses on simulating fractal growth and coalescence of plants with kinetic growth characteristics, yet integration with physiological and ecological models of plants remains a challenge, detracting from the realism of the simulation. The Billboard technique [14] utilizes simple models with appropriate textures to showcase complex objects, but it is limited to two dimensions and cannot realistically account for three-dimensional motion. Finally, voxel space-based approaches [15] have been employed, which consider the concept of space, as well as any reference axis techniques [16] and other modeling methods. With the advancement of deep learning, modeling methods such as 3D-GAN, PointNet, VoxelNet, VAE, etc. have gradually emerged. However, these approaches still pose challenges due to the need for extensive and diverse training datasets, and there is room for improvement in terms of controllability and computational complexity.

Catkin fluttering is a common natural phenomenon, and accurately modeling its flutter is crucial for evaluating the effectiveness of simulation systems. This makes it an essential research component in creating virtual plant models. The application of deep learning methods to simulate catkin shapes encounters challenges stemming from data collection difficulties and the computational complexity involved. However, due to catkins' significant level of self-similarity, describing shapes using the L-system language [17] is programmatically easy to implement, because it is simple in definition, highly structured, and quickly and intuitively simulates plant topologies. Here, we employ the L-system modeling approach to develop an algorithm for generating a 3D catkin model as the fundamental unit of the catkin flutter system.

### 2.2. Dynamic Simulation of Natural Scenery

In recent years, the simulation of natural scenes, such as rain, snow, clouds, smoke, and fire, has become a significant and challenging problem in computer graphics with real-world applications. In 1983, Reeves [18] proposed the stochastic particle system as a

method to simulate complex natural scenes. This approach offers excellent randomness and dynamics, enabling detailed and realistic modeling of objects that undergo continuous changes. Sims [19] later used a parallel particle system to simulate a waterfall and flame, while Loke et al. [20] designed a Particle System Rendering Engine that could generate various pyrotechnic effects, including dazzle. Stam et al. [21] developed a three-dimensional model that captures the ignition, combustion, and extinction of flames based on the flame propagation model using fluid dynamics theory. They presented a flame propagation model that uses the diffusion equation. Meanwhile, Unbescheiden et al. [22] established a particle system based on physical principles to simulate clouds and used texture mapping to draw them. Wang et al. [23] used fluid dynamics principles instead of a particle system to simulate a snow scene. This approach allowed for the real-time simulation of snow scenes and offered a more realistic simulation of the wind and snow interaction.

While significant progress has been made in simulating natural scenes, the majority of research efforts have been concentrated on rain and snow simulations, leaving other areas, such as the fluttering of catkins, relatively unexplored, despite their practical importance. Although rain, snow, and catkins are all suspended solids in the atmosphere, they differ significantly in physical properties and movement behavior. Raindrops and snowflakes possess unique shapes, masses, and falling velocities, while also being influenced by air currents. In contrast, catkins are lighter, slenderer, and primarily influenced by gentle winds and airflow. When simulating catkins' flight, it is crucial to consider their unique shape, mass, and motion characteristics. Additionally, the collision and accumulation behavior of catkins require thorough modeling, which cannot be directly applied to previous simulation methods for rain and snow.

We summarize the experiences gained from simulating natural scenery and present a real-time simulation of catkin fluttering using a combination of particle systems and aerodynamics. By examining the fundamental characteristics of particles and the actual requirements for catkin control, we develop a model that simulates the collision, adhesion, and accumulation of catkin fluttering.

### 2.3. Wind Field Modeling Techniques

The wind is a ubiquitous force of nature that is essential for creating vivid and realistic simulations of natural scenes. Over the years, various wind field modeling techniques have been proposed in the academic community. For instance, WejChert et al. [24] developed an aerodynamics-based wind field model to simulate the flight of leaves in the wind, but this is computationally intensive. Similarly, Wei et al. [25] created a wind field model based on LBGK to simulate the interaction between feathers and the wind field. Zhang et al. [26] explored different wind field simulation techniques, including pulsating wind field simulation and near-surface wind field modeling. In another study, Moeslund et al. [27] simulated the effect of snowflakes flying in the wind based on the N-S equation, but it is also computationally intensive. Wang [28] developed a wind field based on the Boltzmann equation to simulate the interaction of wind and snow fluttering, deposition, erosion, and other phenomena, leading to an incredibly realistic simulation. Additionally, Zhang et al. [29] created a wind field model based on lattice BGK, improved the modeling method of 3D wind field with the *D3Q7* model, introduced the concept of turbulence intensity to describe the randomness of wind, and improved the realism of the simulation.

In order to achieve a realistic wind field simulation and enhance the overall realism of the simulation, we employ the Boltzmann equation as the foundation for simulating the wind field. The D3Q15 model is utilized to track the position of catkins and update their position, velocity, and acceleration in real-time as the wind field changes.

This model facilitates the simulation of the wavering motion characteristics exhibited by catkins during fluttering, effectively capturing their responses to dynamic wind conditions. Considering the influence of the external environment on the motion of catkins, we analyze the force situation of catkins in the process of fluttering and consider the motion

of catkins as a combination of wind motion and free fall, as well as the force of attraction between catkins, so as to build the motion model.

To achieve a realistic simulation of wind fields and improve the accuracy of the simulation, we utilize the Boltzmann equation to simulate the wind field. The *D3Q15* model is employed to track the next position of the catkins, and the position, velocity, and acceleration of the catkin particles are updated in real-time with changes in the wind field to showcase the erratic motion of the catkin during the fluttering process. To account for the influence of external factors on catkin motion, the forces acting on the catkin are analyzed during fluttering, and the motion of catkins is driven by the combination of wind flow, free fall, and the attractive forces between the catkin particles. Based on this, we establish a comprehensive motion model for the catkins.

## 3. Static Model of the Catkin

The fluttering catkins are actually the seeds and derivatives of female poplar or willow trees. Each spring, in order to propagate the next generation, poplar or willow trees release these white, catkin hairs, which carry the seeds and scatter them across the sky with the aid of the wind. To accurately replicate catkins, it is crucial to carefully consider their form, and we employ a fractal-based method to create a three-dimensional representation of catkins.

By analyzing the real-world morphology of the catkin, illustrated in Figure 2a, it becomes evident that it consists of a concise stalk at the top, surrounded by numerous filamentous catkin hairs. This structure can be approximated as shown in Figure 2b, where the brown oval represents the short stalk and point D denotes the starting point of hair growth, from which multiple catkin hairs sprout. Any two hairs in the figure are chosen to draw the structure schematic. For instance, points $D_1$, $D_2$, and $D_3$ are the growth endpoints of node 1, node 2, and node 3, respectively, on the right side. To control the fluffiness of the entire catkin, the first section is rotated around the coordinate axis by an angle $\theta$, whereas the second and third sections are rotated by a slightly smaller angle $\gamma$ to fine-tune the bending of each segment. The rotation angles can be derived from the generative rule by executing the L-system code to determine which axis to rotate around each time.

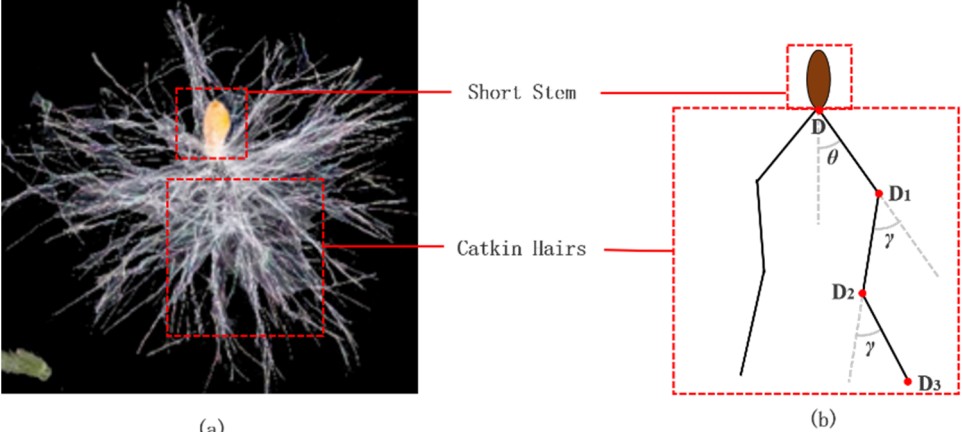

(a)  (b)

**Figure 2.** Simulation structure of catkin. (**a**) Catkin in the real world; (**b**) schematic diagram of the simulated structure of a catkin. The red dots in the figure represent the growth endpoints of each segment on each catkins, while the red lines represent the correspondence between the real structure and abstract structure of the catkins.

Based on the above simulation structure, a three-dimensional model of the catkin is created. The modeling process initiates with the creation of a short stalk at the top of the catkin, approximated as an ellipsoid. First, the y-coordinate axis of the sphere is scaled by a factor of 2 in the right-hand coordinate system. Then, the sphere is drawn to obtain the ellipsoidal model. To ensure that the scaling of the y-axis does not affect the generation of subsequent catkins, a function is required to eliminate the influence of the

previous transformation on the current transformation. This ensures that the subsequent transformations are based on the original, unscaled y-axis.

To render the fine details of catkin hairs, we employ a fractal L-system modeling technique. The catkin's overarching structure displays characteristics of self-similarity and symmetry, reminiscent of a plant's root system. The L-system utilizes an axiom and an iterative substitution rule, with the initial string being the core of the L-system. This generative formula is then iteratively applied to replace the initial string, partially or completely. Each character in the resulting string is assigned a geometric meaning to express complex geometric objects.

To implement the L-system drawing method for the catkin, we need to define the branches using the symbols "[" and "]" in the string. However, unlike other plants, the catkin structure does not have a clear main branch, and instead splits from the root. Moreover, we need to draw multiple hairs from the short stalk of the catkin, and each hair operation must be enclosed in a pair of "[]" to return to the initial state after drawing. The initial state of the hair dispersion is positioned at the bottom of the stalk in the positive direction of the y-axis.

To ensure the correct orientation of catkin hairs, an initial rotation is necessary to reorient the direction of the y-axis. In this study, we introduce a two-step rotation approach, wherein the initial step involves rotation around either the x-axis or the z-axis, followed by a rotation around the y-axis. Figure 3 illustrates the generation process of the catkin hair, exemplified by the first node of the catkin fluff. In Figure 3a, after rotating the hair $OD_1$ clockwise or counterclockwise around the z-axis by an angle $\theta$, multiple hairs, including $OD_1'$, $OD_2'$, $OD_3'$, $OD_4'$, are produced on the xoy plane. In Figure 3b, $OD_1'$ is selected for further study. After rotating it clockwise around the z-axis by an angle $\theta$, then rotating clockwise or counterclockwise around the y-axis by an angle R, and finally advancing by one step, multiple hairs such as $OD_1''$, $OD_2''$, $OD_3''$ are produced in the positive y-axis region above the xoz plane. Similarly, rotating $OD_2'$, $OD_3'$, $OD_4'$, and so on, around the y-axis generates new hair angles, resulting in complete coverage of the spatial 3D area with catkin hair. Table 1 shows the generative rules that assign geometric meaning to each character in the string based on the shape of the catkin.

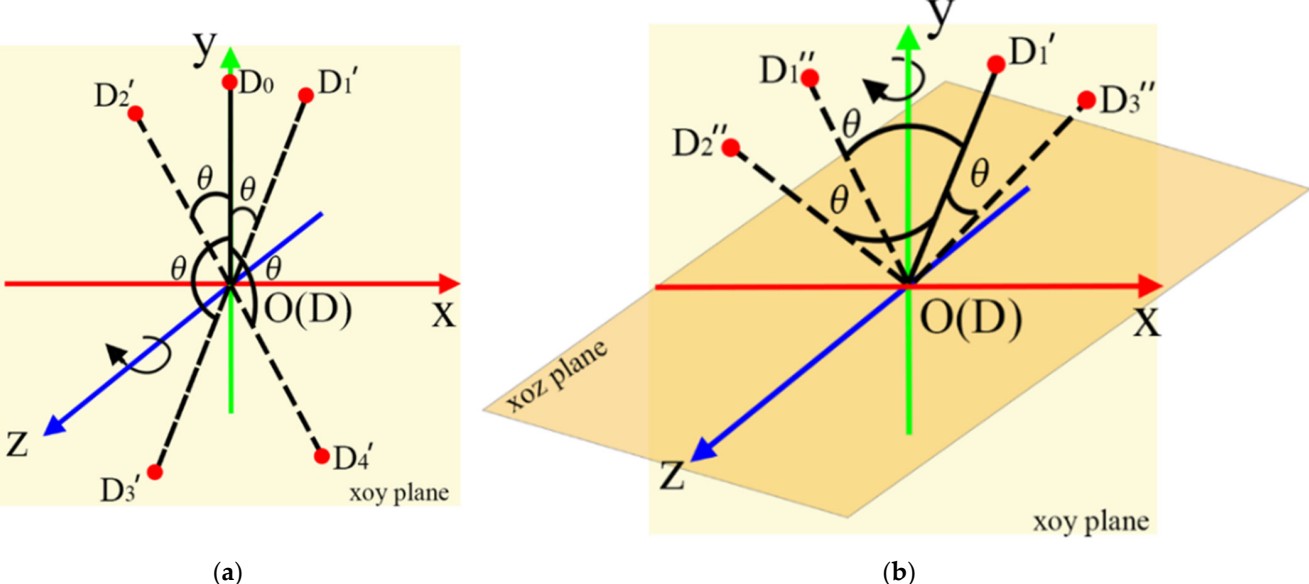

**(a)**　　　　　　　　　　　　　　　　　　　　　　　**(b)**

**Figure 3.** Principle of catkin hairs generation. (**a**) Segment 1 villi first rotate around the x-axis; (**b**) Segment 1 villi then rotate around the y-axis. The light yellow area in the figure represents the xoy plane, while the orange area represents the xoz plane.

**Table 1.** Generative rules.

| Functions | Characters | Meanings |
|---|---|---|
| Forward | *F* | Advance a distance |
| Control the fluffiness of the catkin | \$ | Rotate the angle $\theta$ clockwise around the y-axis |
| | % | Rotate the angle $\theta$ counterclockwise around the y-axis |
| | + | Rotate the angle $\theta$ clockwise around the x-axis |
| | & | Rotate the angle $\theta$ counterclockwise around the x-axis |
| | * | Rotate the angle $\theta$ clockwise around the z-axis |
| | \ | Rotate the angle $\theta$ counterclockwise around the z-axis |
| Control the degree ofbending of the catkin | ? | Rotate the angle $\gamma$ clockwise around the y-axis |
| | . | Rotate the angle $\gamma$ counterclockwise around the y-axis |
| | — | Rotate the angle $\gamma$ clockwise around the x-axis |
| | # | Rotate the angle $\gamma$ counterclockwise around the x-axis |
| | @ | Rotate the angle $\gamma$ clockwise around the z-axis |
| | ! | Rotate the angle $\gamma$ counterclockwise around the z-axis |
| Return to initial state | [ | Putting current state on the stack |
| | ] | Fetching the top-of-stack state |

As shown in Table 1, the fluffiness of the catkin hair is controlled using six characters, such as "\$", while the degree of fluffiness is determined by the maximum angle of rotation on the x and y axes. Similarly, to achieve bending and intertwining of the catkin hair, "?" and the other six characters are used to control the bending degree. The catkin is segmented into multiple sections, and symbols are redefined to accommodate the deflection angle of each segment. This redefinition allows for precise control over the bending angle, contributing to a detailed representation of the catkin's form and structure.

We define the following L-system code to simulate the structure of a catkin:

$$w: \ 50 \times [*\$P] \ 50 \times [*\%Q] \tag{1}$$

$$p: \begin{array}{l} P \rightarrow F@F!F; \ P \rightarrow F@F - F; \ P \rightarrow F - F\#F; \\ Q \rightarrow F@F!F; \ Q \rightarrow F@F - F; \ Q \rightarrow F - F\#F; \end{array} \tag{2}$$

In the equation above, the initial string "*w*" represents the system's original state, while the generative character '*p*' illustrates how the initial string "*w*" extends and rotates around both the y-axis and z-axis to initiate branch growth in each iteration. Even if there is no trunk, the branching process can proceed near the root as usual. The core idea is to introduce randomness in the angles to create various forms of catkin, rather than relying on a variety of syntax. For instance, only the initial strings "[*\$P]" and "[*\%Q]" are used 50 times each. Each time these characters are recognized, the rotation angles obtained from "*", "\$", and "%" characters are different, resulting in filaments distributed into different shapes of catkin hair. Since there is no regularity in the bending of the catkin, the system randomly selects one of the three grammars corresponding to each of them, to be implemented every time "*P*" or "*Q*" is read. By adjusting the length, thickness, number, and fluffiness of the catkin hair, a variety of realistic catkin models can be obtained.

## 4. Dynamic Model of Catkin

Traditional methods of simulating natural scenes involve introducing external forces into the kinetic equations of fluttering catkins to simulate the effect of wind, resulting in simulations that are less realistic [30]. To create a more realistic simulation of the physical motion of catkins and immerse them into the wind flow field, we establish a wind field model based on aerodynamics that adheres to the physical motion law of catkins. We also consider the comprehensive effects of free fall and inter-catkin attraction on catkins and provide a motion model, collision detection and adhesion rules, attraction and stacking mechanisms. By simplifying the acceleration, we are able to create a lifelike depiction

of fluttering catkins in diverse scenarios, considering variations in size, wind speeds, and directions.

*4.1. Motion Model*

4.1.1. Force Model of Catkin in Wind Field

In the real world, the catkin is a small, dense, and light object that is subject to airflow. The effect of drag on its motion is negligible, and the motion of the catkin can be viewed as a combination of wind and free fall, as well as the effect of attraction between catkins. In addition, considering the seasonality of catkins' flying, the difficulty in capturing and measuring data, and limitations in computing resources, we simplify the motion model and believe that gravity, wind, and the attraction of the catkin to a single catkin cluster are the primary factors that affect the motion of the catkin, as illustrated in Figure 4.

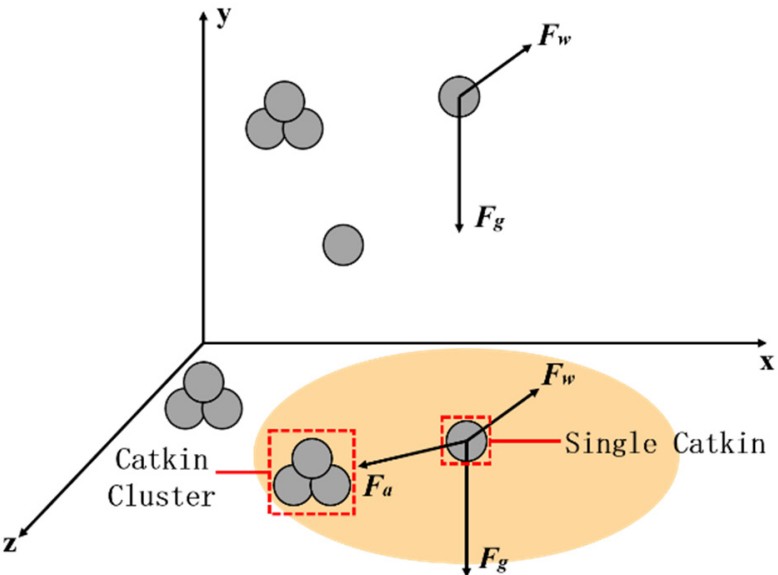

**Figure 4.** Forces on catkins. The orange area in the figure represents the range of attraction of catkins clusters to individual catkins.

In Figure 4, $F_w$ represents the wind force, $F_g$ represents the gravitational force acting on the catkins, and $F_a$ represents the attraction of the catkin cluster to individual catkins. Only the catkin cluster located in the orange area in the figure can attract individual catkins. $F_g$ is the force on the catkin in the vertical direction, primarily affecting its falling speed. On the other hand, $F_w$ and $F_a$ primarily affect the horizontal motion of the catkin, causing it to move back and forth and left and right, while adding variability to the fluttering process. $F_a$ exists only in the following three cases: (1) both the catkin cluster and single catkin are on the ground; (2) both the catkin cluster and single catkin have collided with the wall; (3) the catkin cluster is at an angle between the ground and the wall, and the single catkin is in the air. These forces collaboratively contribute to the intricate motion of the catkins during fluttering, encompassing both vertical and horizontal movements. This integration adds diversity and realism to the simulation, capturing the nuanced dynamics of catkin behavior in response to external influences.

4.1.2. Wind Field Simulation Based on Boltzmann Equation

The *D3Q15* grid model based on Boltzmann's equation [28] is employed to discretize space and time to build a three-dimensional wind field. The Boltzmann equation is a physical equation at the molecular level, providing an accurate mathematical framework for simulating gas molecules in wind fields. In addition, the classic aerodynamic continuous N-S equation is a lower order approximation of the Boltzmann equation, so our discrete wind field modeling method can fully simulate the continuous changes in the wind field

realistically, thereby truly reflecting the flying process of catkins. This model discretizes space and time by dividing space into a grid, where each grid point houses a specific number of wind particles, with each having 15 distinct directions. These wind particles move in their respective directions to neighboring grid points in the next moment, and during this motion they undergo collisions. After collision, the number of wind particles in each direction is redistributed.

1.  Theoretical basis of wind field modeling

Utilizing the *D3Q15* grid to simulate the wind field, we first divide the space into a grid and then sample the grid points in a finite number of directions using distribution functions. Next, we calculate the movement of particles at these points in different directions to simulate the real wind field effect.

The distribution function $f_i(r,t)$ represents the particle density at position $r$ and time $t$ moving in the direction $e_i$. The changes in particle density in different directions reflect the macroscopic distribution and movement of the wind field. The wind field's fluid velocity $u$ and macroscopic density $\rho$ can be expressed using the distribution function as [31]:

$$\rho = \sum_{i=0}^{n} f_i(r,t), \ u = \frac{1}{\rho}\sum_{i=0}^{n} f_i(r,t)e_i \tag{3}$$

To simplify the solution of the Lattice Boltzmann Method (LBM), we replace the non-linear collision term with a linear BGK collision term. The discrete form of the Boltzmann equation can be expressed as [32]:

$$f_i(x + \vec{c_i}\Delta t, t + \Delta t) - f_i(x,t) = \frac{1}{\tau}(f_i^{eq}(x,t) - f_i(x,t)) \tag{4}$$

Here, $f_i(x,t)$ represents the velocity of motion at position x and moment t with respect to $\vec{c_i}$. The collision term is represented by $\frac{1}{\tau}(f_i^{eq}(x,t) - f_i(x,t))$, where $f_i^{eq}(x,t)$ is the particle equilibrium distribution function. $\Delta t$ represents the time step and $\tau$ denotes the relaxation factor. The nonlinearity of the collision term is replaced with a linear BGK collision term to simplify the solution of the LBM.

2.  Modeling of 3D wind field

To better capture the stochastic variation in the real wind field, it is essential to discretize space and time and develop a 3D wind field model using the *D3Q15* approach. The three-dimensional space is discretized into x × y × z grids as depicted in Figure 5. In Figure 5, the grid points in the upper, left and right, lower boundaries of the wind field are denoted as $P_1$, $P_2$, and $P_3$, respectively, while a randomly selected grid point inside the wind field is referred to as $P_4$. To simulate the catkin motion under the effect of gravity and wind, catkins are added to the wind field model. At the $P_4$ grid point, the *D3Q15* model is employed to track the catkins' movement. The possible directions for catkins at the next moment include front, back, left, right, up, down, and six diagonal lines, as well as keeping them stationary. This approach achieves a more precise representation of the wind field's characteristics and behavior.

In the discrete wind field, the distribution of wind particles at each grid point is represented by $F_i(r,t)$, $r$ represents the grid point, and $t$ represents the time. Wind particles can move along i directions, $F_i$ is the fluid density moving along i directions, $\vec{c_i}$ represents the velocity direction of wind particle motion, and the direction of wind particles at each grid point of the *D3Q15* model is shown in Equation (5), where $c$ is the scale parameter associated with the lattice size.

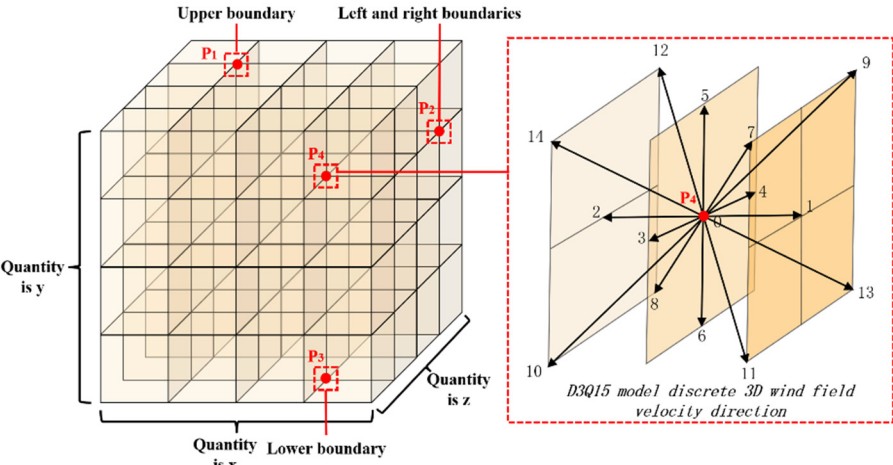

**Figure 5.** Wind field dispersion. The numbers 0–14 in the figure indicate 15 velocity directions on each grid point.

We use $F_i(r, t)$ to represent the distribution of wind particles at each grid point in the discrete wind field, where $r$ denotes the grid point and t denotes the time. Wind particles can move in $i$ different directions, and $F_i$ represents the fluid density of particles moving in the *i-th* direction. The velocity direction of wind particle motion is denoted by $\vec{c_i}$, and the direction of wind particles at each grid point is specified by the *D3Q15* model. The scale parameter associated with the lattice size is denoted by $c$ [33], and the directions of wind particles in the *D3Q15* model are given by Equation (5):

$$\vec{c_i} = \begin{cases} (0,0,0) & i = 0 \\ (\pm 1, 0, 0)c, (0, \pm 1, 0)c, (0, 0, \pm 1)c & i = 1, 2, \ldots, 6 \\ (\pm 1, \pm 1, \pm 1)c & i = 7, 8, \ldots, 14 \end{cases} \tag{5}$$

The density $\rho$ and the velocity field $u$ of the wind particles at each grid point are as follows:

$$\rho(\text{x}, t) = \sum_{i=0}^{14} F_i, u = \frac{1}{\rho} \sum_{i=0}^{14} F_i \vec{c_i} \tag{6}$$

Similar to Equation (4), the dynamics of the three-dimensional discrete wind field are given as:

$$F_i(r + \vec{c_i}, t + \Delta t) = F_i(r, t) + \frac{1}{\tau}(F_i^{eq}(u(x, t), \rho(r, t)) - F_i(r, t)) \tag{7}$$

where $\tau$ is the relaxation factor and $F_i^{eq}(\rho, u)$ is the local equilibrium distribution, which can be obtained from the following equation [34]:

$$F_i^{eq}(\rho, u) = w_i \rho \left[ 1 + \frac{\vec{c_i} \cdot \text{u}}{c_s^2} + \frac{(\vec{c_i} \cdot \text{u})^2}{2c_s^4} - \frac{u^2}{2c_s^2} \right] \tag{8}$$

where $c_s$ takes the value $\sqrt{1/3}$ in the *D3Q15* model and $w_i$ denotes the weight, i.e., the weighting factor of the particle distribution, which can be expressed in the *D3Q15* model as shown in Equation (9):

$$w_i = \begin{cases} 2/9 & i = 0 \\ 1/9 & i = 1, 2, \ldots, 6 \\ 1/72 & i = 7, 8, \ldots, 14 \end{cases} \tag{9}$$

To improve the stability of the wind field, we establish the following boundary conditions [35]. (i)Upper boundary: The upper boundary is represented by the sky in the simulated scene, as shown in Figure 5 at point $P_1$. The vertical velocity is set to 0, while the horizontal velocity is unrestricted. (ii) Left and right boundaries: The wind field's sides

are treated as open boundaries. As depicted in Figure 5 at point $P_2$, the wind's velocity is unrestricted, and it attains equilibrium velocity over an extended period. (iii) Lower boundary: The ground is the lower boundary in the simulated scene, and the momentum of all particles on this boundary's grid point is always 0. This boundary is configured as a bounce boundary, as exemplified at point P3 in Figure 5, where the momenta of all moving particles on each grid point along this boundary are reversed.

3.  Update of the catkin position

Based on the above formula, once the wind speed $u(r,t)$ is calculated for each grid point, it can be utilized to update the position of the catkins. Assuming that the catkins that do not fulfill the attraction condition are affected by both wind and gravity, they move in conjunction with the wind speed, $u(r,t)$, in the wind field and generate a vertical velocity of $-u_g$ due to the effect of gravity. The movement of the catkins at grid point $r_a(x_a, y_a, z_a)$ to $r_b(x_b, y_b, z_b)$ after time $\Delta t$ is expressed as:

$$\vec{c_i} = \begin{cases} x_b = x_a + u_x \Delta t \\ y_b = y_a + (u_y - u_g) \Delta t \\ z_b = z_a + u_z \Delta t \end{cases} \tag{10}$$

Since the end point $r_b$ of the motion may not be situated at a grid point of the three-dimensional wind field, the simulation is approximated via successive interpolation. The presence is assumed of eight 3D grid points around $r_b$, denoted by $P_i$ ($i$ = 0, 1, ..., 7), and the wind field velocity at $P_i$ is represented by $u_i$ ($i$ = 0, 1, ..., 7). As shown in Figure 6, if $r_b$ is not placed exactly on the grid point of the wind field, the linear interpolation method is used to determine the wind field velocity $u_b$ at the location of $r_b$.

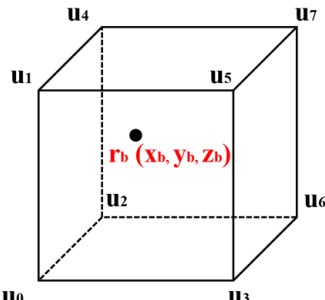

**Figure 6.** Continuous interpolation to obtain the wind speed at the location of catkins.

Let the coordinates of $P_i$ be $(x_i, y_i, z_i)$ ($I$ = 0, 1, ..., 7). We define $\Delta x = \frac{(x - x_0)}{\Delta D_x}$, $\Delta y = \frac{(y - y_0)}{\Delta D_y}$, $\Delta z = \frac{(z - z_0)}{\Delta D_z}$, where $\Delta D_x$, $\Delta D_y$, $\Delta D_z$ denote the lengths of adjacent grids in the x, y, and z directions, respectively. We can then obtain the wind speed at position $r_b$ using the following formula [23]:

$$\begin{aligned} u_b = \quad &(1 - \Delta x)(1 - \Delta y)(1 - \Delta z) \cdot u_0 + (1 - \Delta x)(1 - \Delta y) \cdot \Delta z \cdot u_1 + \\ &(1 - \Delta x) \cdot \Delta y \cdot (1 - \Delta z) \cdot u_2 + \Delta x \cdot (1 - \Delta y)(1 - \Delta z) \cdot u_3 + \\ &(1 - \Delta x) \cdot \Delta y \cdot \Delta z \cdot u_4 + (1 - \Delta y) \cdot \Delta x \cdot \Delta z \cdot u_5 + \\ &(1 - \Delta z) \cdot \Delta x \cdot \Delta y \cdot u_6 + \Delta x \cdot \Delta y \cdot \Delta z \cdot u_7 \end{aligned} \tag{11}$$

Here, $u_i$ represents the wind field velocity at $P_i$, and $\Delta x$, $\Delta y$, $\Delta z$ are used to interpolate the wind speed at the position of $r_b$, where (x,y,z) are the coordinates of $r_b$, and $x_0$, $y_0$, and $z_0$ are the coordinates of the grid point closest to $r_b$.

The equation for calculating the wind speed of catkins (Equation (11)) can be used in combination with Equation (10) to update the position of the catkin and simulate the fluttering process in the wind field. Assigning each catkin its unique and independent wind speed, while adapting environmental conditions accordingly, enables a more authentic simulation of catkins' flight behavior. The quantification of the three-dimensional wind field

also makes it easier to adjust the wind conditions according to the changing environment, thus expanding the potential applications of the system.

### 4.2. Catkin Collision and Adhesion

4.2.1. Collision Detection in the Air

1.  Surrounding sphere of catkins

In deformer simulation modeling, the collision detection algorithm is known to be computationally demanding and time-consuming, posing a significant challenge in enhancing the algorithm's efficiency for virtual simulations. Although collision detection can enhance the realism of the simulation by simulating the phenomenon of catkins sticking and piling up, it also has a notable impact on the system's real-time performance. To achieve a balance between realism and efficiency, we propose a new approach that conducts collision detection at regular intervals instead of checking for collisions in every frame of the simulation. This time interval strategy guarantees the simulation's realism while maintaining real-time performance.

In order to reduce computational time for collision detection and enhance simulation efficiency, the catkins are simplified into spheres, despite having more fluff. Since the simplified catkins have a simple shape and are used in a complex environment, the hierarchical wraparound box method is used for collision detection. This method approximates complex objects with larger enclosing boxes that have simple geometric features [36]. The geometric structure of the catkin is uniformly distributed on all three axes in its abstracted form. Therefore, the catkin is approximated by an enclosing sphere, as shown in Figure 7a. The sphere's radius R is determined by finding the point $D_3$ that is farthest from the starting point D of the catkin fractal and measuring the distance between it and the starting point.

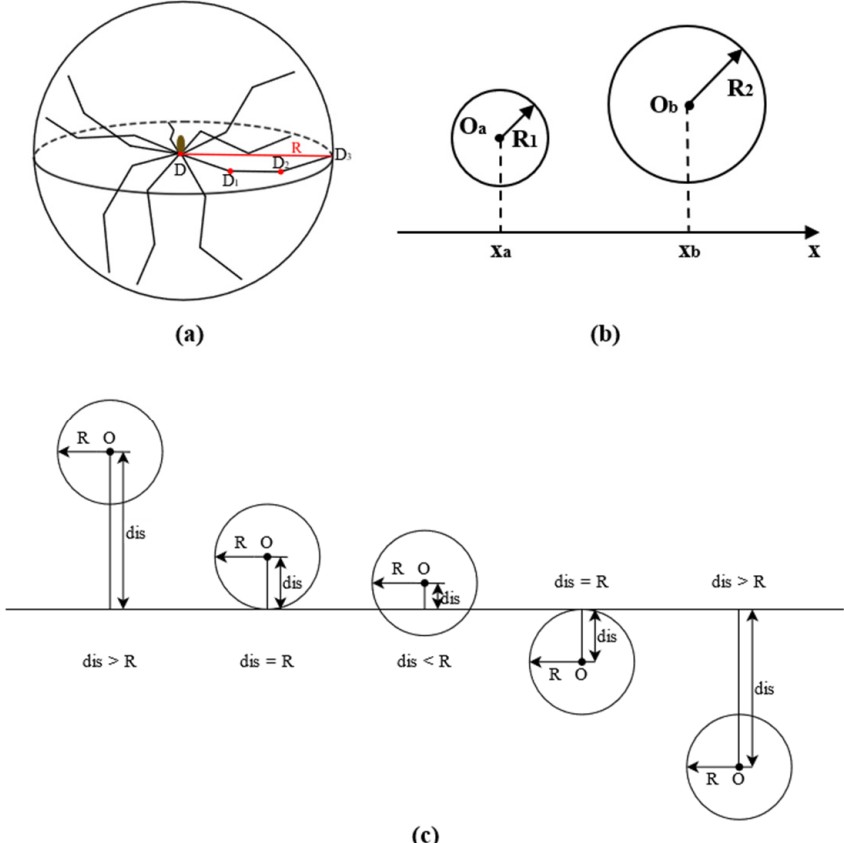

**Figure 7.** Surrounding sphere collision detection principle. (**a**) Surrounding sphere of catkins; (**b**) collision detection of catkins in the air; (**c**) the position of catkins in relation to the ground.

2. Collision detection in the air

Figure 7b illustrates the collision detection process for fluttering catkins. The centers of the two enclosing spheres are denoted by $O_a$ and $O_b$, and their projections on the x-axis are denoted by $X_a$ and $X_b$, respectively. The radii of the two spheres are represented by $R_1$ and $R_2$. If the distance between $O_a$ and $O_b$ is less than or equal to the sum of $R_1$ and $R_2$, then the two catkin spheres are considered to be colliding.

3. Airborne collision response and catkin adhesion

Following a collision, two catkins are considered to be stuck together and move with a common velocity $v_t$, which is simply the average of their pre-collision velocities $v_1$ and $v_2$. Assuming masses of $m_1$ and $m_2$ for the colliding catkins, the conservation of momentum gives us the following equation without considering the contact time or deformation during the collision:

$$m_1 \times v_1 + m_2 \times v_2 = (m_1 + m_2) \times v_t \tag{12}$$

Here, $v_1$ and $v_2$ represent the pre-collision velocities of the respective catkins, while $v_t$ is the post-collision velocity of the two stuck-together catkins. The simplified $v_t$ is given by:

$$v_t = (v_1 + v_2)/2 \tag{13}$$

After the collision, the two catkins move synchronously in the wind field, sharing identical speed, mass, force area, and volume. This synchronized movement creates the appearance of the catkins being stuck together, with their acceleration also being identical.

### 4.2.2. Collision Detection with the Floor or Wall

1. Collision detection with the floor or wall

The simulation currently does not account for the contact time and deformation of catkins during a collision. When catkins collide with walls, they follow the principle of conservation of momentum. The position relationship between the catkins and the ground can be divided into five cases, which are illustrated in Figure 7c. The center of the enclosing ball of catkin is represented by O, the radius of the enclosing ball is denoted by R, and the distance between the center of the ball and the ground is represented by dis. Since collision detection occurs at regular intervals, the catkin may be on the ground at one moment with a distance from the ground that is less than or equal to R, while in the next moment, it may be beneath the ground with a distance from the ground that is also greater than R. Thus, there is a possibility that the catkin penetrates the ground. Taking into account the aforementioned position relationship, the collision between the catkin and the ground can be classified into two main categories: (1) on the ground and the distance from the ground is less than or equal to the radius of the enclosing sphere; and (2) under the ground. Similarly, the collision between catkins and walls can be categorized in a similar way to the ground.

Furthermore, special attention should be paid to the collision detection conditions of catkins that are adhered to each other in the air. When these catkins adhere to one another, they create a catkin cluster characterized by identical velocity and acceleration. Consequently, collisions involving the cluster and the wall or ground differ from those involving individual catkins. In this case, the collision detection conditions need to be modified as follows. (1) For ground collision detection, identify the catkin with the lowest y-coordinate in the catkin cluster and check if it has collided with the ground, using the previously mentioned single catkin collision detection method. (2) For ground collision response, move the catkin that collided with the ground to a position tangent to the ground, record the displacement of this catkin, update the y-coordinates of all particles in the cluster by the same displacement, and mark them as collided with the ground. Similarly, the collision detection and response for the wall is similar to that of the ground, except that the catkin searched for is the one with the highest x-coordinate in the catkin cluster.

2.    Response to collision with the floor or wall and catkin adhesion

When responding to a collision with the ground, the catkin should be moved to a position that is just tangential to the ground. If the catkin is attached to other catkins, then the vertical velocity of all the catkins it is attached to will be reduced to 0, and their vertical acceleration will also be reduced to 0.

Similarly, when responding to a collision with the wall, the catkin should be repositioned to a point that is just tangent to the wall. If the catkin is attached to other catkins, then the horizontal velocity of all the catkins it is attached to will be reduced to 0, and their horizontal acceleration will also be reduced to 0.

*4.3. Catkin Attraction and Accumulation*

The catkin, being small and lightweight, frequently experiences collisions with other catkins as it drifts, resulting in the form of clusters. Catkin fibers have a high lignin content, ranging from 19 to 31%, which makes them excellent at adsorption. Moreover, the hair on each catkin in the cluster is charged by rubbing against each other and can attract light objects, such as individual catkins, leading to accumulation. We consider three cases where individual catkins accumulate due to the attraction of catkin clusters: (1) both the clusters and the individual catkins are on the ground; (2) both the clusters and the individual catkins have collided with walls; (3) the catkin clusters are at an angle between the ground and the wall, and the individual catkins are in the air.

In our method, based on statistical distribution, we believe in the existence of an attractive force between the catkin cluster and a single catkin, which increases as the distance decreases. Drawing inspiration from the Smoothed Particle Hydrodynamics (SPH) method, we introduce a Gaussian kernel function to simulate this attraction. The Gaussian kernel function (RBF) uses a Gaussian distribution (also known as a normal distribution) as the attenuation function. The decay rate of the control function is exponential and is determined by the Euclidean distance and $\gamma$ parameters. The Gaussian distribution is extensively employed in statistical analysis due to its statistical rationality, smoothness, symmetry, and adherence to the central limit theorem. Its applicability and descriptive power make it suitable for calculating the attraction among catkins in our method. To calculate the attraction of individual particles to catkins, we have adapted the Gaussian kernel function (RBF) and designed equations, as shown in Equation (14):

$$F(\vec{x_i}, \vec{x_j}) = e^{-\gamma \|\vec{x_i} - \vec{x_j}\|^2} \tag{14}$$

In Equation (14), the attraction value between the catkin cluster and individual catkin is determined by a parameter $\gamma$ greater than zero, as well as the distance between the catkin cluster and the catkin coordinates. As the distance decreases, the attraction value approaches 1, and as the distance increases it approaches 0. We have illustrated the relationship between the attraction value (Fa) and distance (D) in Figure 8. The attraction calculation function exhibits a bell-shaped curve with smoothness and symmetry, while the curve reaches its peak at the mean D = 0 and gradually decreases to both sides. This curve shows the behavior of the attraction calculation function, providing a clearer understanding of the attractive force between the catkin cluster and individual catkins. Moreover, it substantiates the statistical rationality of utilizing the Gaussian function to simulate the attraction between catkins and individual particles.

By utilizing the aforementioned function, we are able to calculate the attraction force F of the catkin cluster to the single catkin. As the attraction force decreases with the increasing distance between the catkin cluster and the single catkin, the acceleration of the single catkin intensifies while it is being drawn closer and moved by the attraction, resulting in a variable acceleration of linear motion. However, due to the complexity of this motion, it is divided into multiple uniformly accelerated linear motions in order to calculate the displacement. We analyze the relative motion of a single catkin to the catkin cluster, recalculating the attraction force, velocity, acceleration, and displacement after each

time step until the distance between the catkin cluster and the single catkin is less than a certain value. At this point, it can be considered that a single catkin has been attracted to the position of the catkin cluster, resulting in a pile-up and concluding the attraction process.

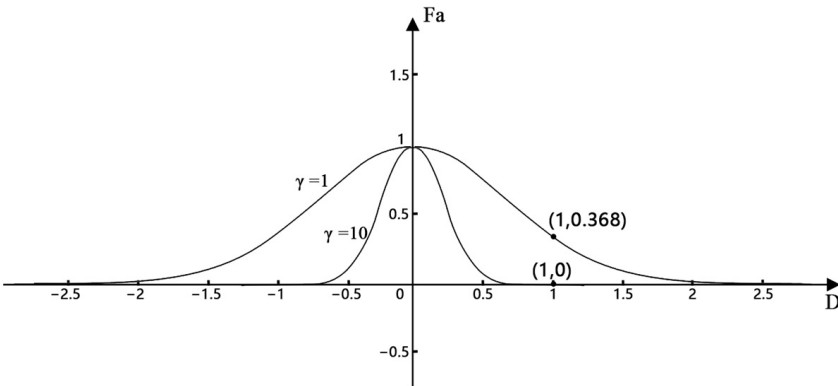

**Figure 8.** Attraction calculation function.

The process of attraction between the catkin cluster and a single particle is illustrated in Figure 9. The distance between the catkin cluster and the single catkin is denoted as $dis_0$, $dis_1$, $dis_2$, $dis_3$, which can be calculated from the position coordinates. The attractive forces of the catkin cluster to the single catkin are denoted as $F_0$, $F_1$, $F_2$, $F_3$, which are obtained by substituting $dis_0$, $dis_1$, $dis_2$, $dis_3$ into Equation (14), respectively, and increase with decreasing distance. The proportionality coefficients $P_0$, $P_1$, and $P_2$, are obtained by dividing the attractive forces $F_0$, $F_1$, and $F_2$ by the mass of the catkin, which is approximated as 1 due to the small mass of the catkin. The displacement of a single catkin after each time step is denoted as $\Delta dis_1$, $\Delta dis_2$, $\Delta dis_3$, which are obtained by multiplying the distance between the catkin cluster and a single catkin by the proportionality coefficients $P_0$, $P_1$, and $P_2$, respectively. As the distance decreases, the gravitational force becomes larger, and the acceleration becomes greater, leading to an increasing displacement of a single catkin in each time step in the order of the scale factor from $P_0$ to $P_2$ of the total distance between the current catkin cluster and a single catkin. The stacking phenomenon can be simulated by multiple attractions between catkins.

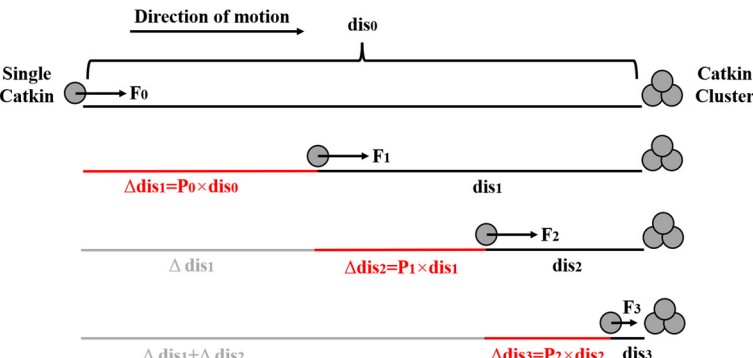

**Figure 9.** The attraction process of a catkin cluster to a single catkin.

## 5. Results and Discussion

### 5.1. Experiment Platform

The experiments were conducted on an Intel(R) Core(TM) i7-10875H CPU @ 2.30GHz processor, NAVIDIA GeForce RTX 2060 graphics card with 16.0 GB of RAM (15.9 GB available), running on a 64-bit Windows OS with an x64-based processor. The simulation system was built using the VC++ development platform and the OpenGL graphics library.

### 5.2. Catkin Modeling Results

Figure 10a shows the modeling process for catkins. Our method allows for adjustments to the length, thickness, number, and fluffiness of the catkin, while also including bent and intertwined hairs to enhance visual appeal and achieve a more realistic scene. To simulate different forms of catkins, each feather's rotation angle in the model can be individually controlled using the hyperparameters $(\theta, \gamma)$. Figure 10b shows the modeling results when using the same range of hyperparameters $(\theta, \gamma)$; the catkin models will be different in details rather than identical. Because each segment of each catkin hair is generated by bending it at a random angle in a random direction, the fluffiness and bending angle of the catkin hairs in the five models are not the same, ensuring the diversity of the models. This diversity contributes to a realistic representation of catkin structures with distinct characteristics. Figure 10c shows five representative catkin models that have been imported into the simulation system. Each catkin is composed of three sections, and the angle of rotation of Section 1 around the x-axis, y-axis, and z-axis is denoted by $\theta$, while $\gamma$ represents the rotation angle around Sections 2 and 3 of the catkin around the x-axis, y-axis, and z-axis. By adjusting these rotation angles, it is possible to model catkins with varying fluffiness and curvature.

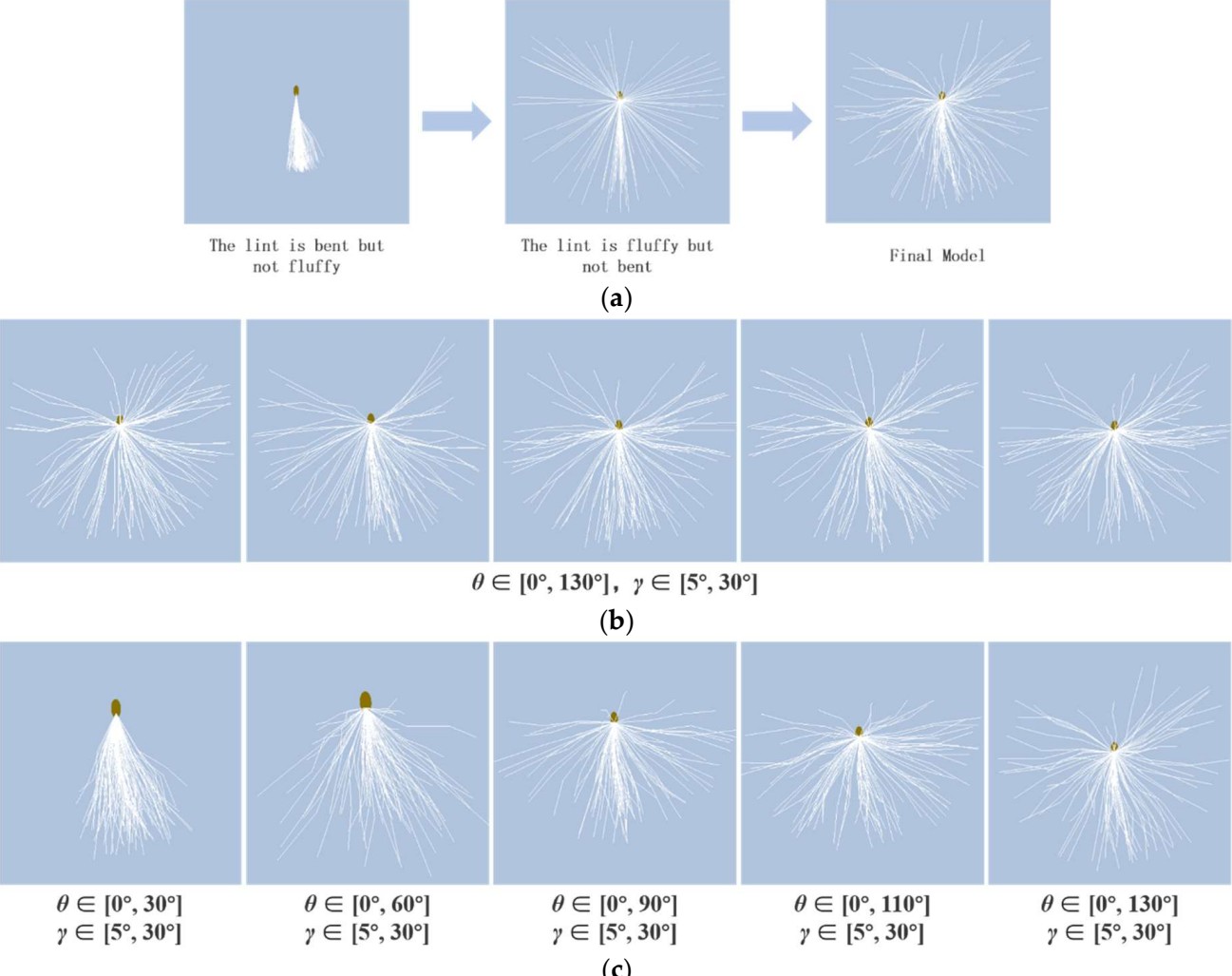

**Figure 10.** Modeling process diagram of catkins. (**a**) Catkin modeling process; (**b**) catkin modeling results under the same parameters; (**c**) catkin modeling results under different parameters.

Figure 11 shows the visual comparison between the modeling effect of catkins and real-world catkin photos, with (a) shows the real effect, (b) shows the modeling effect, and

(c) partially magnifies the last two images of (a,b) with many catkins to display the details of the catkin model. It can be seen that the catkin modeling method based on the L-system proposed in this paper can truly restore the catkins' shape, providing strong support for simulating the catkins' flying process.

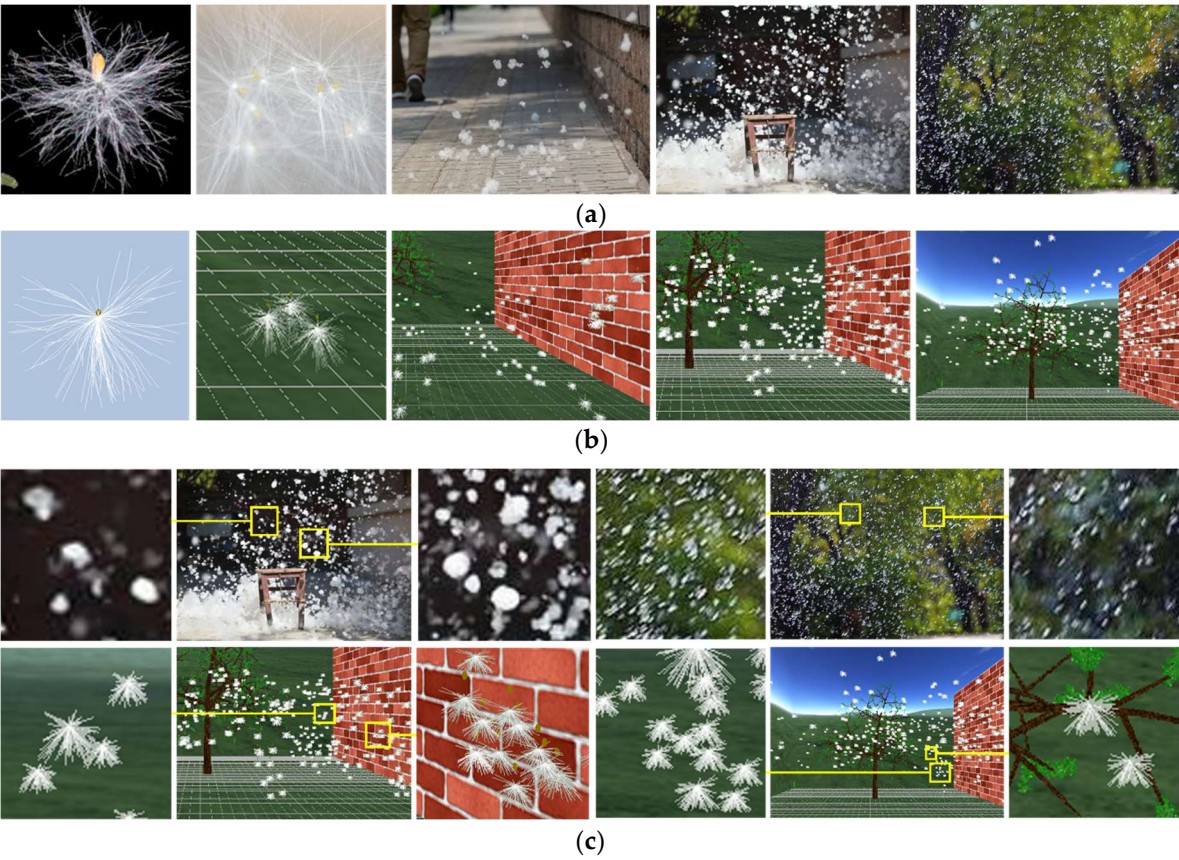

**Figure 11.** Comparison between simulated and real effects. (**a**) Realistic effects; (**b**) simulation effects; (**c**) comparison of modeling details.

To validate our simulation results, a total of 489 valid questionnaires were collected. Appendix A shows questionnaires filled out by ordinary users, while Appendix B shows questionnaires filled out by forestry experts. The respondent ratio between ordinary individuals and forestry experts was maintained at 3:2. Ordinary people contributed generalized opinions and evaluations, while forestry experts focused on identifying professional issues and offering suggestions. In addition, weighted statistical analysis methods were employed to address potential biases resulting from imbalanced sample proportions. Specifically, different weights were assigned to the two groups based on the actual proportion of ordinary people and forestry experts. Each ordinary person received a weight of 2, and each forestry expert received a weight of 3, reflecting their relative importance in the population. During statistical analysis, these weights were utilized to ensure a balanced consideration of both ordinary individuals and forestry experts' perspectives in the results. This weighted statistical analysis enhanced our ability to accurately estimate the characteristics and distribution of viewpoints in the overall sample, rectifying potential biases due to proportional imbalances.

Statistical analysis demonstrated that, compared to the five sets of images depicted in Figure 11a, over 71% of respondents evaluated the simulation effect of Figure 11b as acceptable, realistic, and very realistic on average. Particularly noteworthy was the second group of control images, where over 80% of users rated the simulation results as acceptable, realistic, and even more realistic. The comparison between the five sets

of images in Figure 11a,b suggests that our simulated results exhibit a certain level of authenticity, effectively portraying the static structure and dynamic process of catkins in flight. To ensure the result credibility, we sought input from experts in relevant fields and obtained valuable insights from an ecological perspective on simulating the phenomenon of flying catkins. Subsequently, we made corresponding adjustments to the shape and simulation effect of catkins. From the experimental results, it becomes evident that the L-system modeling method employed in our study surpasses traditional approaches to natural scenery modeling, including Billboard technology [14], triangle face combination methods, texture mapping [37], and other two-dimensional modeling techniques. Notably, our L-system method fulfills the demands for three-dimensional motion more effectively. In contrast to modeling techniques, like deep learning and crystal growth [38], our approach stands out for its simplicity and efficiency. Additionally, it offers the advantage of easily fine-tuning model details, allowing us to achieve more realistic modeling results.

*5.3. Catkin Fluttering Process Simulation Results*

The application of computer technology to simulate natural phenomena has yielded powerful tools for scientific research, leaving a profound impact on various fields, such as engineering, medicine, and design. Numerous scholars have successfully simulated complex three-dimensional natural scenes, including phenomena like flames and rain, while the most relevant work to our method is the simulation of snow. For example, Masselot et al. [39] and Langer et al. [40] analyzed the drifting process of snowflakes under wind transmission in a two-dimensional scenario. Additionally, Bryan et al. [2], Saltvik et al. [41], and Xu et al. [42] addressed challenges related to realism and speed in rendering snowy scenes.

While the simulation of snow has inspired our work, distinctions exist between catkins and snowflakes concerning their shape, falling method, and lightness. As a result, the methodologies applied in simulating snow cannot be directly transposed. First, the irregular shape of catkins may manifest as a slenderer or sheet-like structure, rendering the direct application of mathematical models similar to those for snowflakes complex. Furthermore, catkins, being lighter and softer, exhibit increased susceptibility to air resistance, causing them to fall in a relatively stable manner.

By utilizing computer technology, we are able to realistically recreate the serene scene of catkins fluttering in the sky through simulation. This simulation involves the creation of the skybox, walls, and other environmental elements using VC++ and OpenGL. Parameters related to wind field strength and catkin count are adjusted to simulate the natural visual case of catkin fluttering. Initially, catkins are generated and scattered randomly throughout the sky using random functions. Then, their speed, acceleration, and position are altered based on the wind field's propagation and the effects of gravity and catkin attraction. At each time step, assessments are made to determine whether catkins are colliding, sticking, attracting, or stacking, and their movements are adjusted accordingly. Derived from our experimental results, our simulated catkins display a heightened level of dynamism and grace in comparison to snowflakes as they gracefully traverse through the air. Particularly noteworthy is their agile and lightweight nature throughout the entire flight process, demonstrating an impressive capability to sustain a stable descent even when influenced by varying wind currents.

When simulating the process of catkins flying, the wind field environmental parameters set in this paper are shown in Table 2. Here, $S_w$ represents the size of the wind field, which is a three-dimensional array. The x, y, and z coordinates represent the number of grids that the wind field divides into in the x, y, and z directions. $N_w$ represents the number of wind particles initially distributed at each grid point. g represents gravitational acceleration, which is a three-dimensional array. $\Delta t$ represents the time step size and updates the position of flying particles according to the time step size. $D_w$ represents the wind direction in the wind field, which is a three-dimensional array. This paper sets the wind direction to be consistent with the positive x-axis direction. $L_w$ and $V_w$ represent wind level and

average wind speed, respectively. After searching for relevant data, they are set to 3 and 2 based on empirical values. $V_m$ represents the maximum velocity at the boundary, with a value of 1000. These parameters collectively define the wind field and the conditions under which catkins are simulated to flutter.

**Table 2.** Wind Farm Environmental Parameters Table.

| Parameter Names | Parameter Values |
|---|---|
| $S_w$ | (70, 70, 70) |
| $N_w$ | 50 |
| g | (0, −0.8, 0) |
| $\Delta t$(s) | 0.3 |
| $D_w$ | (1, 0, 0) |
| $L_w$ | 3 |
| $V_w$ (m/s) | 2 |
| $V_m$ (m/s) | 1000 |

The flying process of catkins is simulated using the parameter values in Table 2. Figure 12 illustrates the physical motion of 100 catkins in the simulation. Figure 12a,d demonstrates the realistic simulation of catkins fluttering, colliding, and adhering in the air. The drift process of the catkin is randomized, resulting in a realistic simulation effect. Figure 12b,e simulate the fluttering of catkins colliding with the ground, accumulating on the surface, and adhering to it. Through controlled adjustments to the viscosity of the ground, we can effectively manage the fluttering of catkins, preventing them from being easily blown away by the wind. Figure 12c,f simulate the effect of catkins colliding with the wall, accumulating at the corner, and being blocked by the wall. By changing the height of the wall, the movement of catkins can be obstructed, providing a realistic reference for catkin control. These simulation results empower us to study and gain a deeper understanding of catkin behavior in diverse environmental scenarios. Moreover, they provide valuable insights into potential control mechanisms that can be implemented to regulate the fluttering and accumulation of catkins in real-world environments, including gardens and urban landscapes.

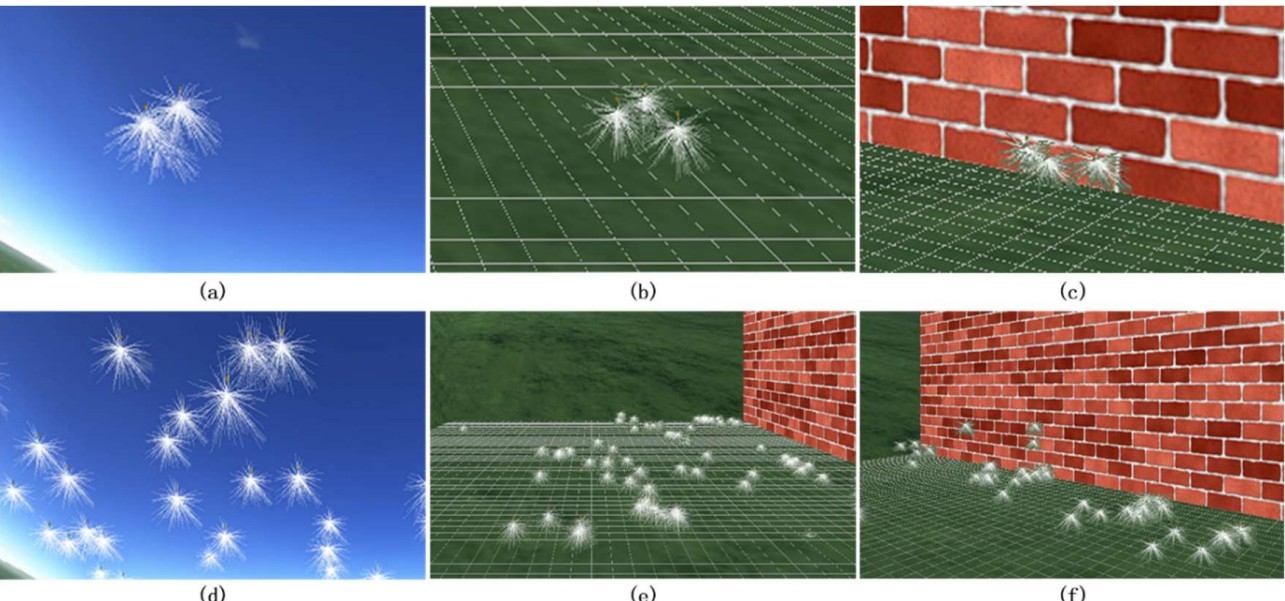

**Figure 12.** Catkins physical movement simulation effects. (**a**) Sticking in the air; (**b**) accumulation on the ground; (**c**) pile up at the foot of the wall; (**d**) floating in the air; (**e**) stuck to the ground; (**f**) blocked by walls.

By adjusting the number of catkins, three sizes of catkins can be simulated: large, medium, and small. Several sets of experiments were conducted to simulate the catkin movement, and some of the results are given in Figure 13. The catkin model is a detailed 3D model, with each catkin consisting of 100 hairs and involving 400 vertices. This complexity increases the rendering time and reduces the frame rate. As illustrated in Figure 13a, a small number of catkins (500) results in a sparse distribution; as illustrated in Figure 13b, an appropriate number (1000) resembles the amount of catkins in daily life; as illustrated in Figure 13c, a large number (2000) can simulate the clustering of catkins in large scenes, but the system overhead is too high, and the frame rate is too low, which compromises the real-time performance of the system. The experimental results validate the effectiveness and feasibility of our approach, which can simulate the fluttering, collision, adhesion, attraction, and accumulation of catkins, resulting in realistic simulations. These experiments also highlight the influence of adjusting the number of catkins on system performance. This can guide developers in optimizing simulations for both realism and efficiency, ensuring that simulations remain interactive and responsive in real-time applications.

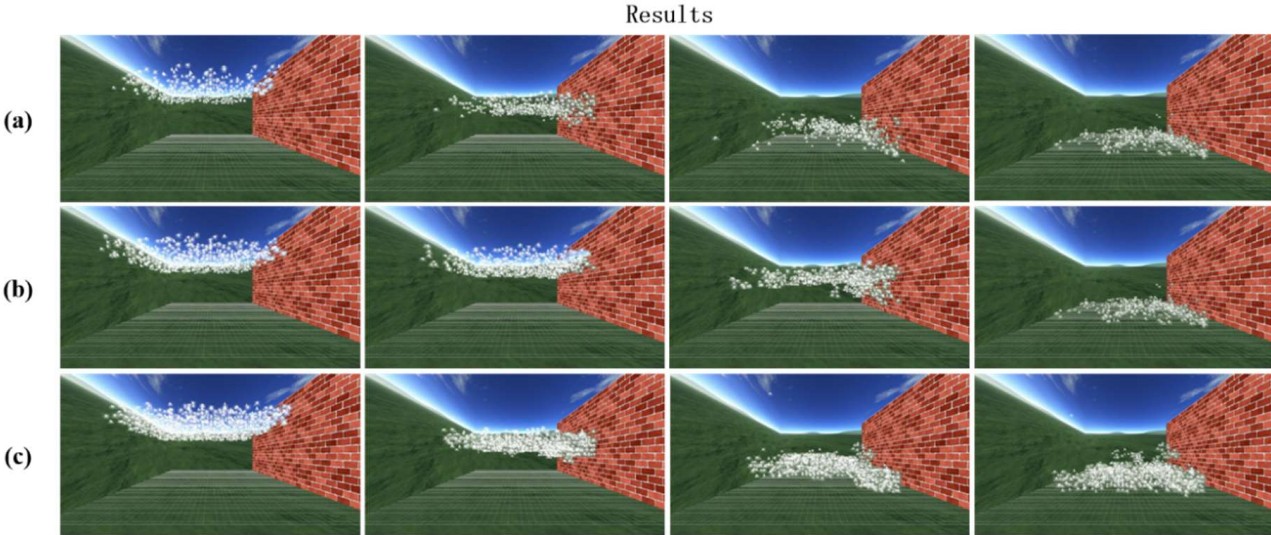

**Figure 13.** Catkin fluttering simulation effects. (**a**) Simulation effect at 500 catkins; (**b**) simulation effect at 1000 catkins; (**c**) simulation effect at 2000 catkins.

Due to the complexity of the structure of catkins, a large number of point and line structures are needed to represent their morphology. In this paper, each catkin model consists of 400 vertices and 100 lines. Table 3 shows the data statistics of the number of points, lines, and simulation time that need to be processed for updating all flying particles during the simulation process. It can be seen from Table 3 that, with the increase in particle numbers in catkins, the number of points and lines to be drawn increases significantly, which brings the cost of significantly slowing down the simulation speed.

**Table 3.** Statistical table for simulation data for flying catkins.

| Catkins No. | Points No. | Lines No. | Simulation Time (s) | Simulation Effect |
|---|---|---|---|---|
| 500 | 200,000 | 50,000 | 1.234 | Figure 13a |
| 1000 | 400,000 | 100,000 | 1.625 | Figure 13b |
| 2000 | 800,000 | 200,000 | 5.438 | Figure 13c |

*5.4. Comparative Experiment*

In the realm of modeling catkins, there exists a substantial time efficiency gap between leveraging computer technologies, such as L-system modeling, and manual modeling using 3D software, like 3ds Max 2016 and Blender 3.5.

Our choice is the L-system for catkin modeling, a method that swiftly generates catkin morphology and intricate structures by specifying simple rules. The iterative and fractal nature of L-systems makes it highly efficient, enabling the creation of numerous catkin models in a relatively short timeframe. Additionally, L-system modeling offers the advantage of automatic generation, eliminating the need for manual modeling one by one and significantly enhancing time efficiency.

On the contrary, using 3D software, like 3ds Max and Blender, for catkin modeling involves manual modeling and fine adjustments, necessitating individual simulation of the shape, details, and dynamic properties of catkins. While 3ds Max and Blender pro-vide greater flexibility and accuracy, this approach demands more time and effort, especially when dealing with a large quantity of catkins.

Table 4 gives a time comparison for catkin modeling using different methods, considering a single catkin model composed of 400 vertices and 100 lines. The modeling time using 3ds Max and Blender in Table 4 is an estimated value contingent on the modeler's proficiency and the model's accuracy. As shown in Table 4, when constructing a catkin model of the same scale, the L-system method employed in this paper iteratively generates the catkin model using computer interpretation code, saving significant time and effort and improving efficiency to dozens of times that of manual modeling. Moreover, given the requirement for simulating the flight process of catkins with numerous particles of varying shapes in this study, the L-system method proves more suitable for catkin modeling than manual modeling.

**Table 4.** Comparison of modeling time for catkins.

| Method | Catkins No. | Points No. | Lines No. | Simulation Time (s) |
|---|---|---|---|---|
| 3ds Max | 1 | 400 | 100 | 2400 |
| | 5 | 2000 | 500 | 10,000 |
| blender | 1 | 400 | 100 | 1800 |
| | 5 | 2000 | 500 | 7500 |
| L-system | 1 | 400 | 100 | 3 |
| | 5 | 2000 | 500 | 15 |

## 6. Conclusions

We propose a physics-based method that models and dynamically simulates the process of catkin fluttering. By utilizing aerodynamics to mimic the natural phenomenon of catkins, our method can guide the control of catkin fluttering and has broad application prospects. The main results of our method are as follows. (1) The rendering of catkins is achieved. We create a catkin particle system based on natural morphology and physical characteristics to achieve three-dimensional realistic modeling of catkins. (2) The Boltzmann equation is used to simulate the wind field, and the *D3Q15* model is employed to discretize three-dimensional space. The motion of the catkin is considered as a result of both the wind's influence and the integrated effects of free fall and catkin inter-attraction. (3) The characteristics of the catkin are also incorporated, and the collision and adhesion, attraction, and accumulation of the catkin fluttering process are further simulated to enhance the realism of the catkin fluttering scene. In summary, this paper designs and implements a catkin simulation system, and the experimental results validate that our method can accurately and effectively complete the simulation of the catkin fluttering process and achieve a realistic natural effect in 3D scenes, improving efficiency by dozens of times compared to manual modeling. The statistics from the survey questionnaire indicate that over 71% of users provided acceptable, authentic, and more realistic evaluations of the simulation results, thereby validating the authenticity of our simulation outcomes. This study has potential value in multiple fields: firstly, the modeling and flying process of catkins not only focuses on their behavior, but also is closely related to meteorology and environmental protection, with potential impacts on urban environment, air quality, and human health; Secondly, research has revealed that flying catkins may have adverse

effects on air quality, especially for asthmatic and allergic patients, thus helping to improve air quality management strategies; in addition, the flying process of catkins may also have an impact on wind energy equipment and agriculture. Further research on catkin simulation will provide more information about the interaction between catkins; finally, this study provides an important case for interdisciplinary research, emphasizing the importance of applying physical modeling and simulation techniques to solve practical environmental problems.

Future work includes using GPU acceleration to improve the efficiency by transferring the updated calculation of wind field particles to the GPU, as well as improving the generality of our system in order for it to be easily applied to simulation of other natural phenomena.

**Author Contributions:** Conceptualization, J.Z. and M.Y.; methodology, J.Z. and M.Y.; software, J.Z.; validation, J.Z., M.Y. and W.M.; formal analysis, J.Z.; investigation, J.Z.; resources, J.Z., B.X., J.D. and Q.H.; data curation, J.Z.; writing—original draft preparation, J.Z.; writing—review and editing, J.Z., M.Y. and W.M.; visualization, J.Z.; supervision, M.Y. and W.M.; project administration, J.Z. and Q.H.; funding acquisition, B.X., J.D. and W.M. All authors have read and agreed to the published version of the manuscript.

**Funding:** This research was supported in part by the National Natural Science Foundation of China (Nos. 32271983, U22B2034, 6226070885, 62102414, 62172416, 61972459, 62002358), in part by the Open Project Program of State Key Laboratory of Virtual Reality Technology and Systems, Beihang University (No. VRLAB2023B01), and in part by the Personalized Viewing System for the Museum Project of College Students' innovation and entrepreneurship training program, Beijing Forestry University (No. X202110022192).

**Data Availability Statement:** The datasets analysed during the current study are not publicly available due to ethical and privacy restrictions but are available from the corresponding author on reasonable request.

**Acknowledgments:** The authors thank students from Beijing Forestry University, including Xilin Jiang, Jiachen Xue, Benrun Zhang, Xiaoxuan Zhang, Yefan Li, Shuhan He, Yan Li, You Wu, Chunying Hu, Qi Cui, and Feixiang Huang, for their contributions to this article. We also thank the anonymous reviewers who made valuable comments which helped to improve the manuscript.

**Conflicts of Interest:** The authors declare no conflict of interest.

## Appendix A

General survey questionnaire on physically based modeling and dynamic simulation evaluation of fluff flying process.

Dear survey participants,

In order to further improve the physically based modeling and dynamic simulation of catkins, we are conducting a questionnaire to collect subjective evaluations of the modeling and simulation effects in this work. We kindly invite you to provide answers that align with your expectations based on the catkins' 3D model, the effects of the catkins' floating process, and your personal impressions. Please rest assured that your personal information will be kept strictly confidential. Thank you for taking the time to participate!

1. Personal information

(1) Your age:
   ○ 18–25 years old        ○ 26–35 years old        ○ 36–45 years old
   ○ 46–55 years old        ○ 56 years old and above

(2) Your educational background:
   ○ Junior high school       ○ High school             ○ College/Undergrad
   ○ Master's degree or above   ○ Other (please specify)_________

(3) Your major:
   ○ Computer related        ○ Forestry related       ○ Others

(4) Do you understand the modeling of catkins, the process of catkins flying, or visual simulation:
   ○ Yes                      ○ No

2. Realistic evaluation of catkins' 3d modeling

(1) Please evaluate the overall authenticity of the computer simulated catkins model (the three images in group (**a**) are real photos, and the three images in group (**b**) are simulation images):

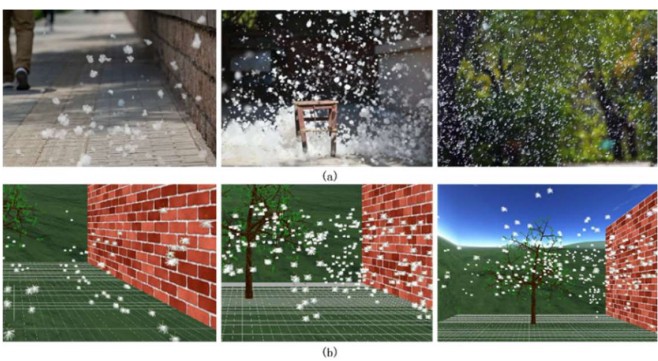

○ Very real     ○ Real     ○ Commonly     ○ Unreal     ○ Very

(2) Please evaluate the authenticity of the details of the computer simulated catkins model (the two images in group (**a**) are real photos, and the two images in group (**b**) are simulation images):

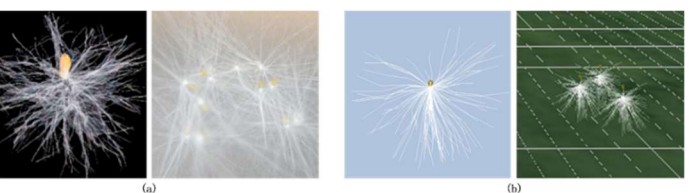

○ Very real     ○ Real     ○ Commonly     ○ Unreal     ○ Very

(3) What aspects do you think have affected the realism of Catkins' 3D modeling?

○ The shape and size of catkins
○ The shape and size of catkins
○ Details of the background environment
○ Other (please specify)_________

(4) If you believe that the visual effect is not realistic enough or completely untrue, please provide your specific opinions or improvement suggestions (optional):

Your suggestion: _________________

(5) Do you think the visual effects of computer simulated catkins models can help demonstrate the propagation characteristics of catkins?

○ Computer related     ○ No     ○ Not sure

3. Realistic evaluation of the flying process of catkins

(1) Please evaluate the realism of the computer-simulated catkins floating process (with 500, 1000, and 2000 catkin particles in groups (**a**–**c**) respectively):

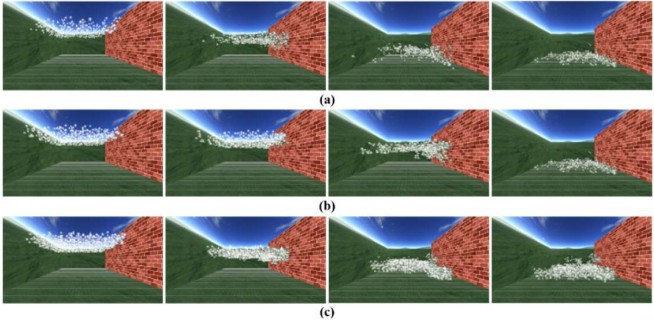

○ Very real     ○ Real     ○ Commonly     ○ Unreal     ○ Very

(2) What aspects do you think affect the realism of simulating the flying process of catkins?

○ Flying direction     ○ Flying speed
○ Flying density     ○ Other (please specify)_________

(3) If you believe that the visual effect is not realistic enough or completely untrue, please provide your specific opinions or improvement suggestions (optional):

Your suggestion: ___________________

(4) Do you think the visual effects of computer simulation of the flying process of catkins can help demonstrate the propagation characteristics of catkins?

○ Computer related          ○ No          ○ Not sure

## Appendix B

Expert survey questionnaire on physically based modeling and dynamic simulation evaluation of fluff flying process.

Dear survey participants,

In order to further improve the physically based modeling and dynamic simulation of catkins, we are conducting a questionnaire to collect subjective evaluations of the modeling and simulation effects in this work. We kindly invite you to provide answers that align with your expectations based on the catkins' 3D model, the effects of the catkins' floating process, and your personal impressions. Please rest assured that your personal information will be kept strictly confidential. Thank you for taking the time to participate!

1. Personal information

(1) Your age:

○ 18–25 years old          ○ 26–35 years old          ○ 36–45 years old

○ 46–55 years old          ○ 56 years old and above

(2) Your educational background:

○ Undergraduate          ○ Master's degree          ○ Doctoral degree

(3) Your major:

○ Computer related          ○ Forestry related          ○ Others

(4) Do you understand the modeling of catkins, the process of catkins flying, or visual simulation:

○ Yes          ○ No

2. Realistic evaluation of catkins' 3d modeling

(1) Please evaluate the overall authenticity of the computer simulated catkins model (the three images in group (**a**) are real photos, and the three images in group (**b**) are simulation images):

○ Very real          ○ Real          ○ Commonly          ○ Unreal          ○ Very

(2) Please evaluate the authenticity of the details of the computer simulated catkins model (the two images in group (**a**) are real photos, and the two images in group (**b**) are simulation images):

○ Very real          ○ Real          ○ Commonly          ○ Unreal          ○ Very

(3) What aspects do you think have affected the realism of Catkins' 3D modeling?
○ The shape and size of catkins
○ The shape and size of catkins
○ Details of the background environment
○ Other (please specify)__________

(4) If you believe that the visual effect is not realistic enough or completely untrue, please provide your specific opinions or improvement suggestions (optional):
Your suggestion: ____________________

(5) How much potential do you think computer simulated catkins models have in solving forestry research and management problems?
○ Great　　　　　　　○ To a certain　　　　○ No potential　　　○ Limited

(6) How do you think computer simulated catkins models can assist in forestry planning and management? Please provide your thoughts and suggestions.
Your suggestion: ____________________

3. Realistic evaluation of the flying process of catkins

(1) Please evaluate the realism of the computer-simulated catkins floating process (with 500, 1000, and 2000 catkin particles in groups (**a**–**c**) respectively):

(a)

(b)

(c)

○ Very real　　　　　○ Real　　　　　○ Commonly　　　○ Unreal　　　○ Very

(2) What aspects do you think affect the realism of simulating the flying process of catkins?
○ Flying direction　　○ Flying speed
○ Flying density　　　○ Other (please specify)__________

(3) If you believe that the visual effect is not realistic enough or completely untrue, please provide your specific opinions or improvement suggestions (optional):
Your suggestion: ____________________

(4) What is the potential application of computer simulated catkins flying process in the forestry field?
○ Assessing the impact of willow catkins dispersal on forest
○ Assist in forestry planning and
○ Help predict the path of forest fire
○ Other (please specify)__________

(5) How much potential do you think computer simulation of the flying process of catkins has in solving forestry research and management problems?
○ Great　　　　　　　○ To a certain　　　　○ No potential　　　○ Limited

(6) How do you think computer simulation of the flying process of catkins can assist in forestry planning and management? Please provide your thoughts and suggestions.
Your suggestion: ____________________

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
