# Peer review of "Physics-Based Modeling and Fluttering Dynamic Process Simulation for Catkins"

_forests, doi:10.3390/f14122431_

Round 1

Reviewer 1 Report (Previous Reviewer 1)

Comments and Suggestions for Authors

Please refer to my report for corrections.

Author Response

Reviewer 2 Report (Previous Reviewer 2)

Comments and Suggestions for Authors

The revised manuscript has improve significantly since my last few reviews. The main issue that I keep having is tthat the manuscriopt was written more like a student thesis rather than a research paper. But now it is much better. Just one last point on the "Acknowledgements", please cut down and not turn it into a thank you speech like in your student thesis, especially "I would first like to thank my supervisor , whose expertise was inval- 819 uable in formulating the research questions and methodology. Your insightful feedback 820 pushed me to sharpen my thinking and brought my work to a higher level. In addition, I 821 would like to express my gratitude to Xilin Jiang, Jiachen Xue, Benrun Zhang, Xiaoxuan 822 Zhang, Yefan Li, Shuhan He, Yan Li, You Wu, Chunying Hu, Qi Cui, Feixiang Huang, and 823 other students from Beijing Forestry University for their contributions to this paper. Their 824 research and achievements have provided valuable reference and inspiration for me. Fi- 825 nally, I could not have completed this dissertation without the support of my parents and 826 friends, who provided stimulating discussions as well as happy distractions to rest my 827 mind outside of my research." --- please be more concise. 

Comments on the Quality of English Language

-

Author Response

This manuscript is a resubmission of an earlier submission. The following is a list of the peer review reports and author responses from that submission.

Round 1

Reviewer 1 Report

Comments and Suggestions for Authors

Please refer to the attached report for my detailed suggestions and their rationale.

Comments on the Quality of English Language

Please refer to the attached report for my detailed suggestions and their rationale.

Reviewer 2 Report

Comments and Suggestions for Authors

The authors focus on the simulation of catkin fluttering. The research also delves into the complexities of modeling catkin shapes using deep learning methods and geometric techniques. My reasons are as follows:

1. what is the research objectives and research value of this work? The research primarily focuses on the simulation of catkin fluttering, which is simply too narrow. The manuscript does not offer substantial contributions to the wider field of ecology or environmental protection and lacking of novelty.

2. the reading is very difficult as it reads like a report and appears to be an academic exercise (at best), with very limited research value.

3. "To validate our simulation results, we distributed 500 survey questionnaires and received 352 valid responses. Statistical analysis revealed that, when compared to the sets of images depicted in Figure 11 (a), an average of over 71% of respondents...." --- this is not robust at all. How can you just validate the results by "conducting a survey"? The validation needs to be supported by scientific evidence and logical reasoning.

4. The manuscript delves into deep learning methods and geometric techniques for simulating catkin shapes. However, it does not clearly address the challenges or limitations of these methods, which could be crucial for practical applications. Implementation is also lacking in details. 

5. While the manuscript mentions various modeling methods like 3D-GAN, PointNet, VoxelNet, and VAE, it doesn't provide a robust theoretical framework to support the chosen methodology, potentially making the research less rigorous.

6. Finally, there are simply too many grammatical errors and some are quite basic. like 'challenging task' instead of 'challenge task' just to name one example out of the many issues.

For the above reasons, I am afraid that even a major revision is not sufficient to address the concerns.

Comments on the Quality of English Language

-

Round 2

Reviewer 1 Report

Comments and Suggestions for Authors

I am very satisfied with the revision provided by the authors. I recommend publication as is.

Reviewer 2 Report

Comments and Suggestions for Authors

My opinion has not changed since from the previous round, that the manuscript is not yet ready for publication in its current form. Below are some specific concerns that are still not addressed:

1. The methodology employed in your study lacks the rigor expected for scientific research. Specifically, the ratio of ordinary people to forestry experts among the respondents to your questionnaire is 5:1. This imbalance raises questions about the validity of your findings.

2. While it's commendable that you sought expert opinions for your simulation results, it is very concerning that there are no scientific or technical validations to substantiate the correctness of your work.

3. The sudden inclusion of 7-8 references without clear justification or context is puzzling and detracts from the coherence of the manuscript.

4. I think the research statement is too narrow, the research value is limited.

My opinion is that the manuscript, in its current state, appears to be a work in progress. It resembles a student report that has been repackaged into a manuscript, leaving many gaps unaddressed.

In light of these issues, I cannot recommend the manuscript for publication at this time. 

Comments on the Quality of English Language

-